# Microcystin-LR Removal from Water via Enzymatic Linearization and Ultrafiltration

**DOI:** 10.3390/toxins14040231

**Published:** 2022-03-22

**Authors:** Abelline Fionah, Cannon Hackett, Hazim Aljewari, Laura Brady, Faisal Alqhtani, Isabel C. Escobar, Audie K. Thompson

**Affiliations:** 1Department of Chemistry, University of Kentucky, Lexington, KY 40506, USA; akfionah1@uky.edu; 2Ralph E. Martin Department of Chemical Engineering, University of Arkansas, Fayetteville, AR 72701, USA; cjhacket@uark.edu (C.H.); hsaljewa@uark.edu (H.A.); fhalqhta@uark.edu (F.A.); 3Department of Chemical and Materials Engineering, University of Kentucky, Lexington, KY 40506, USA; laura_brady5@uky.edu (L.B.); isabel.escobar@uky.edu (I.C.E.)

**Keywords:** microcystin, harmful algal blooms, linearization, enzyme, ultrafiltration

## Abstract

Microcystin-LR (MC-LR) is a toxin produced by cyanobacteria that can bloom in freshwater supplies. This study describes a new strategy for remediation of MC-LR that combines linearization of the toxin using microcystinase A, MlrA, enzyme with rejection of linearized byproducts using membrane filtration. The MlrA enzyme was expressed in *Escherichia coli* (*E. coli*) and purified via a His-tag with 95% purity. Additionally, composite membranes made of 95% polysulfone and 5% sulfonated polyether ether ketone (SPEEK) were fabricated and used to filter a solution containing cyclic and linearized MC-LR. Tests were also performed to measure the adsorption and desorption of MC-LR on polysulfone/SPEEK membranes. Liquid chromatography-mass spectrometry (LC-MS) was used to characterize the progress of linearization and removal of MC-LR. Results indicate that the MlrA was successful at linearizing MC-LR. Membrane filtration tests showed rejection of 97% of cyclic MC-LR and virtually all linearized MC-LR, with adsorption to the membranes being the main rejection mechanism. Adsorption/desorption tests indicated that methanol could be used to strip residual MC-LR from membranes to regenerate them. This study demonstrates a novel strategy of remediation of microcystin-tainted water, combining linearization of MC-LR to a low-toxicity byproduct along with removal by membrane filtration.

## 1. Introduction

Microcystin-LR (MC-LR) is among the most problematic algal toxins in America [1], with several outbreaks and appearances in fresh water supplies in recent years. Microcystins are cyclic peptides with seven amino acids, and the congener with leucine (L) at position 2 and arginine (R) at position 4 is known as MC-LR [2]. Various bacteria have been shown to contain an mlr gene cluster for the expression of a protein triad, which consists of MlrA, MlrB, and MlrC, responsible for the degradation of MC-LR [3,4]. The mlr gene cluster has characterized only one biodegradation pathway for microcystin by Sphingomaonas sp. strain ACM-3962 [3]. These individual enzymes cleave the MC-LR toxin at different bond locations in the molecule. MlrA initiates the MlrABC degradation pathway [5] by linearizing MC-leucine-arginine (MC-LR) by a peptide bond cleavage between adda and arginine. This cleavage removes around 60% of MC-LR toxicity [5,6,7,8,9,10]. The expression and purification of MlrA has been attempted numerous times to achieve higher yields and investigate MC degradation. MlrA is one of the most problematic proteins to purify, and some report that MlrA is hard to fold correctly when expressed in a host cell [7]. While looking for a new novel way to enhance MlrA yield, a study done by Jiang et al. found that adding MC-LR to LB media while expressing the enzyme stimulated and increased the protein yield [11]. Wu et al. added L-cysteine and graphene oxide to the MlrA enzyme and extended the lifetime of the enzyme [12]. Liu et al. was the first team to reach more than 90% MlrA purity by adding a maltose-binding protein tag (MBP) [7]. The MBP-tag kept the protein soluble and reduced protein degradation in contrast to His6x-tag, the most common protein tag that can form insoluble and misfolded proteins in *Escherichia coli* (*E. coli*) [13,14].

For the removal of MC-LR from water, biological treatment methods have shown limited use because the microbes require a long residence time to eliminate the toxins [15]. Biofiltration with naturally occurring bacteria has shown to take more than eight days to show signs of MC-LR removal [15]. Photocatalytic removal is most effective at acidic pH, but potency is decreased under alkaline pH [16,17]. Additionally, while ozonation has been proven to be the most effective, its byproducts potentially lead to bacteria growth [15]. Membrane filtration has shown varying levels of success in the removal of MC-LR. In a study by Eke et al., hydrophobic and negatively charged polysulfone (PSf) membranes were found to reject approximately 59% of MC-LR via adsorption to the membrane surface and charged repulsion of the negatively charged MC-LR [15]. The partial success of modified membranes and reactive processes suggests that a physical barrier membrane with additional functionality (e.g., charge, reactivity) may result in enhanced MC-LR remediation.

Polyether ether ketone is a high-performance thermoplastic. Due to its excellent chemical, thermal, and mechanical properties, which are brought about by its aromatic backbone, it has been utilized in many high-end products that require solvent resistance, high thermo-oxidative stability, broad chemical resistance, oxidation stability, and passive biocompatibility [17,18]. Though PEEK has many advantages, it has associated disadvantages, including its insolubility in most organic solvents, rendering PEEK difficult to functionalize, as well as its hydrophobicity nature. One method to address these drawbacks is via structural modifications of PEEK [18]. One way to improve the hydrophilicity of the compound is by introducing charged functional groups into the polymer chain. This can be caried out by introducing sulfonated functional groups into the aromatic back bone, which provides electrons for functionalization [18], more specifically the hydroquinone unit between the two ether bridges, where second-order electrophilic substitution occurs preferentially [19,20]. Introduction of the sulfonated functional groups using sulfuric acid not only increases the hydrophilicity of PEEK, but it also results in a product, sulfonated polyether ether ketone (S-PEEK), that is soluble in polar aprotic solvents [17].

The molecular weight of MC-LR is approximately 1 kDa, while literature molecular weight cutoff (MWCO) ranges for UF and NF membranes are 10 kDa–100 kDa and 1 kDa–10 kDa, respectively [1]; therefore, based on MWCO, MC-LR should not be removed by the process of size exclusion. MC-LR has a weak negative charge and is hydrophilic at pH values of 9.5–10 [2]; therefore, we expected MC-LR would be rejected by hydrophobic and negatively charged membranes due to repulsive charge interactions between the membrane surface and MC-LR [3,4,5]. PSf membranes showed higher rejection in the initial stages because of adsorption of MC-LR on the surface and adsorption of MC-LR to the membrane. However, as the filtration test progressed, the binding sites available to MC-LR for adsorption on the surface reduced, in turn reducing the rejection provided by the membrane. This led to the synthesis of a new membrane via blending of PSf and sulfonated polyether ether ketone (SPEEK) specific with a significantly higher negative charge to increase charge repulsion between the membrane and MC-LR and hence increase rejection.

Furthermore, combining membranes with enzymes is a promising strategy. The work presented here operated the system as a sequential membrane bioreactor (MBR), with enzymes free in water for the destruction of MC-LR and with membranes ensuring the rejection of byproducts. MBRs combine biological and membrane treatment to effectively remove/destroy contaminants of concern. MBRs are similar to conventional activated sludge systems with the exception that the biomass responsible for removing the contaminants of concern are retained within the bioreactor component of the system using membranes rather than secondary clarifiers [6]. Existing MBRs use ultrafiltration membranes (0.1 µm to 0.01 µm pores, retains > 20 kD) to retains solids and cells.

Under mild conditions, SPEEK does not readily form membranes, as it is prone to hydrogel formation [19]. Due to this, SPEEK is often incorporated into a polymer that can form membranes. Polysulfone (PSf) is a resin polymer with high chemical resistance, thermal stability, and applicability to wide ranges of pH [21]. Though PSf has been vastly applied to water treatment, its hydrophobic nature makes it prone to membrane fouling [22]. Introducing SPEEK into its matrix increases the hydrophilicity of this polymer and reduces the rate at which it fouls. The composite material of SPEEK and PSf exhibits an increased chemical and thermal stability as well as increased hydrophilicity, and it has shown great promise in filtration studies [23]. The composite membrane can filter out particles, such as microbes, organic molecules, as well as divalent compounds as ultrafiltration membranes [24]. However, due to the added charge introduced by the SPEEK, charged molecules can also be removed via charge repulsion [25]. Therefore, SPEEK-PSf membranes are expected to filter out charged as well as hydrophobic byproducts.

In this study, we explored the addition of a His-tag to MlrA to purify it. A methodology was optimized to prevent protein insolubility and misfolding while expressing it in *E. coli*. The enzyme was evaluated for purity and activity to linearize MC-LR. Physical treatment has limitation in removing MCs from water, and membrane separation can serve as a barrier. During water treatment, cleaving MC-LR or linearization of MC-LR is a key step in degradation, and eliminating the byproducts are also important. Following linearization, a SPEEK-PSf membrane was investigated to absorb toxin byproducts, and the rejection mechanism was proposed and identified. Membranes were characterized by FTIR, SEM, XPS, and zeta potential. The toxin linearization and removal were evaluated with filtration efficiency characterized by LC-MS.

## 2. Results and Discussion

### 2.1. MlrA Expression and Experiments

#### 2.1.1. Protein Expression and MlrA Purification

*Escherichia coli* (*E. coli*) containing cloned pETMlrA plasmid was expressed in a normal condition after optical density 1 reached ~0.6 units (at 600 nm) by using 0.5 mM IPTG. MlrA chromatogram eluted from the Co-NTA column at 62.5–125 mM imidazole. A sample of the pooled fractions containing MlrA was analyzed by SDS-PAGE, and the results are shown in Figure 1. The results showed that MlrA migrated to a position in the gel close to the 42 kDa as indicated by the marker in lane 1. The MlrA matched the molecular weight that was designed in this study.

The purity of MlrA eluted in 125 mM imidazole was estimated to be about 95%, while less purity of MlrA eluted in 62.5 mM imidazole. This purity assessment confirmed by SDS-PAGE as a general detection of total protein and Western blot as a specific detection for MlrA as His-6x-tag protein. As shown in Figure 1, a single band of MlrA enzyme in 125 mM imidazole was detected using Coomassie brilliant blue staining as well as a His6x-tag detection in Western blot (data not shown). These results proved the full length of MlrA expression and purity.

#### 2.1.2. Microcystin-LR Linearization Activity of MlrA

MC-LR was treated with MlrA in a phosphate buffer (5 mM) solution. LC-MS was used to analyze samples taken at 0, 4, and 24 h after initial mixing (Figure 2). At the start of the experiment, all MC-LR was present in the cyclic form (5.03 min). MS detected a peak for the cyclic MC-LR at *m*/*z* = 995.6 (Appendix A). At 4 h, the cyclic peak was reduced in height, and an additional peak for linear MC-LR was present at 5.17 min. MS detected peaks for the linear MC-LR at *m*/*z* = 1013.8 (+1 charge state) [3,4] and *m*/*z* = 507.4 (+2 charge state). At 24 h, the linear MC-LR peak was clearly visible. Based on single-ion monitoring (SIM) chromatograms (Appendix A), a 58% reduction in the concentration of cyclic MC-LR occurred from 0 to 24 h. Since MlrA has been shown to linearize many MCs (MC-LR, MC-RR, -YR, -LY, -LF, and -LW) and nodularin, this remediation method is applicable to other toxins.

### 2.2. Membrane Fabrication and Experiments

#### 2.2.1. Morphological Characterization

SEM images (FEI Quanta scanning electron microscope, North America) (Figure 3) show the morphology of the surface as well as the cross-section of the SPEEK-PSf membrane before filtration of the byproducts of the enzymatic linearization of MC-LR (Figure 3A,C, respectively) and after filtration (Figure 3B,D, respectively). The images show a unform pore distribution across the surface of the membrane and finger-like projections of the pores in the cross section. A dense layer can be observed in the cross-section images, which was characteristic of the NIPS process indicating the polymer-rich active area. The average pore diameter was found to be 50 nm and characterizes the pore size of the as-synthesized membrane. Particles can be observed on the surface image of the membrane after filtration in Figure 3A. However, the cross-section image of the same membrane (Figure 3C) does not show particles in the pores. This indicated that fouling only took place on the surface of the membrane due to the presence of charge.

#### 2.2.2. Structural Characterization

FT-IR spectra (Figure 4) shows the characteristic peaks in the SPEEK-PSf membrane. The O=S=O peak around 1020 cm^−1^ and 1080 cm^−1^ indicated the presence of the SPEEK in the membrane matrix. Aromatic back bone character was observed around 1490 to 1510 cm^−1^. There was C-H stretch around 2900–3000 cm^−1^. The absence of the carbonyl functional group at around 1650 cm^−1^ corresponding to the backbone carbonyl band in the membrane after filtration indicated that there could be potential fouling taking place that obstructs the peak. These were consistent with previous literature where the membranes were synthesized in the same manner [23].

XPS spectra showed the presence of various elements on the surface of the SPEEK-PSf membrane. As seen in Table 1 and Table 2, more elements were observed on the surface of the membrane after filtration than before filtration, indicating accumulation of particles onto the membrane surface, for example, the appearance of chlorides and phosphorous. Chlorides and phosphorus on the surface of the membrane after filtration can be attributed to the phosphate buffer added to make the MIrA solution during the linearization process. Table 1 and Table 2 showed higher percentage of C, N, O, and S in the membrane before filtration compared to the membrane after filtration. The C1s peak at 284.92 eV was at a higher percentage 71.25% in the membrane spectra before filtration than that after filtration 51.25, which could be attributed to the change in aromaticity when the byproducts were filtered through. Furthermore, there was also a change in the S peaks, with the presence of only one prominent peak at 167 eV before filtration, while there was a second peak in the spectra of the membrane after filtration at ~163 eV. This indicated that there was a reduction of the sulfone function group. Additional changes were observed with respect to the S2p peaks. There was a decrease in the percentage from 1.92% to 0.82% after filtration, which could be attributed to the loss of O functionality. There was a peak at 530.75 eV for O1s in the spectrum of the unfiltered membrane belonging to the sulfonic group. The percentage decreased from 22.08% in the unfiltered membrane to 20.08 eV in the membrane after filtration can be attributed to either the decrease in the O=S=O functional group or the formation of new carbon–oxygen bonds. The small percentage of Na present on the membrane surface before filtration can be attributed to surface contamination during sample preparation. However, the percentage of Na increased after membrane filtration due to the presence of the phosphate buffer in the solution.

#### 2.2.3. Contact Angle and Zeta Potential

The average contact angle of the as-synthesized membrane was determined to be 66.6° ± 1.91°. The low contact angle indicated that the membrane was hydrophilic in nature, which is close to literature values obtained for similar membranes utilizing the same procedure (48.3° ± 0.67°) [23]. Though this was higher than what was reported in literature, it still indicates that the membranes are hydrophilic in nature compared to the control membranes that were synthesized with just PSf/NMP. These membranes had an average contact angle of 72.79° ± 1.66°. The reduction in the water contact angle indicated that there was increased hydrophilicity of the membrane with the addition of SPEEK polymer, which can be associated with the presence of sulfonic groups that also increased the negativity of the membranes. Sulfonic functional groups generate dipole moments onto the membrane surface, increasing both the charge and the hydrophilicity of the membrane [26]. Previous literature using the same procedure indicated that the SPEEK-PSf membranes were negatively charged, as shown in studies by Ike et al. [23], in which zeta potential measurements at pH 6 were found to be −61 ± 4.6 mV.

#### 2.2.4. Adsorption and Desorption of Microcystin-LR on Membranes

Experiments were performed to determine the adsorption and desorption behavior of MC-LR to a SPEEK-PSf membrane. LC-MS was used to analyze samples taken at 0, 4, and 24 h after initial mixing (Figure 5). Note that methanol was added just before the sample for 24 h was taken. At t = 0 h, a cyclic MC-LR peak was visible at 5.06 min. At t = 4 h, the cyclic MC-LR peak was still present, but the concentration was reduced to 27% of the starting amount (based on SIM chromograms, Appendix A), indicating that most of the MC-LR has adsorbed to the membrane. At t = 24 h, after methanol addition, the cyclic MC-LR concentration increased to 73% of the starting amount, indicating that methanol caused most of the adsorbed MC-LR to desorb.

These results were consistent with other studies that found that microcystin adsorbs to various types of plastic, but methanol reduced adsorption. For example, Altaner et al. found that microcystin congeners dissolved in water adsorbed to polypropylene pipette tips, but methanol (≥40%) rectified the adsorption losses [27].

#### 2.2.5. Filtration of MC-LR and Linearization Byproducts through Membrane

For filtration experiments, MC-LR was incubated with MlrA for 24 h, and the resulting solution was passed through a membrane to determine if MC-LR or any byproducts would be adsorbed or rejected by the membrane (Figure 6). For this experiment, a different UPLC column (peptide column) was used than in previously described experiments, so the MC-LR peaks occurred at different times from those shown in Figure 2 and Figure 5, where a C18 column was used. Additionally, single-ion monitoring was used to increase the clarity of the MC-LR peaks.

At t = 0 h (Figure 6A), a cyclic MC-LR peak was visible at 4.63 min. MS detected peaks for the cyclic MC-LR at *m*/*z* = 995.6 and 498.5 (+ 2 ion) (Appendix A). At t = 24 h (Figure 6B), a cyclic MC-LR peak was still visible at 4.63 min, but there was also a linear MC-LR peak at 4.09 min. MS detected peaks for cyclic MC-LR at *m*/*z* = 995.6 and 498.5 (+2 ion) and peaks for linear MC-LR at *m*/*z* = 1013.8 and 507.5 (+2 ion). At t = 24 h, the area of the cyclic MC-LR peak was only modestly reduced, indicating a 11% decline in the concentration of cyclic MC-LR. This could be because of degradation of the MlrA enzyme over the course of long-term storage. The MlrA was stored at −20 °C for approximately two years before being used for this experiment. Loss of MlrA activity can occur over the course of storage. For example, Wu et al. found that MlrA lost approximately 50% of activity in degrading nodularin, a related toxin, after 4 days of storage at 0 °C [12]. MlrA would be expected to lose activity at a slower rate at −20 °C, but the loss could still be significant over the course of long-term storage. Nevertheless, the peak for linear MC-LR at t = 24 h indicated that some activity persisted.

For the (partially) degraded MC-LR solution filtered through the SPEEK-PSf membrane (Figure 6C), the peak for cyclic MC-LR at 4.65 min was much smaller than it was before filtration (97% reduction in peak area). Furthermore, the peak for linear MC-LR was no longer visible. The small peak at 1.73 min is likely a contaminant or impurity, and the peak was not visible in any other samples. The tall curve that escaladed from ~7 min onwards was from the phosphate buffer. This curve is more clearly visible for this sample vs. other samples due to the much shorter *y*-axis scale. In the mass spectrums for this sample, the MC-LR peaks were not distinguishable from background noise due to the significantly reduced concentration of MC-LR.

The LC-MS results of the filtrate indicated that the vast majority (97%) of cyclic MC-LR was adsorbed or rejected by the membrane, and virtually all linear MC-LR was adsorbed or rejected. The membrane could be more selective for linear MC-LR than cyclic MC-LR because the larger surface area of linear MC-LR enabled stronger adsorption to the membrane. The results from the previous section indicated that the membrane could be soaked in methanol solution to desorb attached MC-LR and regenerate the membrane.

TOC measurements were carried out to determine how much total dissolved carbon was filtered out utilizing the SPEEK-PSf membrane (Table 3). After membrane filtration, there was reduced total organic carbon evident in the filtrate. TOC of the stock containing the MC-LR, water, and ethanol was 252.5 mg/L. The TOC after reaction increased due to additives, such as acetone to stop the reaction and MIR-A enzyme, that were added to the solution. It should be noted that the reaction was concentrated up to 4× for filtration studies, which contributed to the high concentrations of the TOC measurements observed. The reduction in the TOC concentration after filtration indicates that the as-synthesized membrane was successful in removing some of the MC-LR enzymatic linearization byproducts.

## 3. Conclusions

Predicted increases in surface water temperatures, combined with current agricultural practices, are expected to result in the continued expansion of algal blooms containing cyanobacteria and enhanced public health risk due to algal toxins migrating into potable water. The most common cyanotoxins released during algal blooms are microcystins (MCs). Due to MCs cyclic peptide structure, removal via conventional treatment processes remains challenging. Enzymatic remediation allows an engineered catalysis system to potentially destroy these toxins. A new treatment strategy of combining MlrA enzymes with SPEEK-PSf ultrafiltration membranes has been shown in this work to be promising. MlrA successfully linearized MC-LR into less toxic byproducts, which were then removed via membrane filtration. In this study, an active MlrA enzyme was produced from a heterologous host, and the enzyme was able to linearize MC-LR and convert it to linearized MC-LR. The linearized MC-LR along with some undegraded cyclic ones were then filtered through composite hydrophobic and highly negative membranes made of 95% polysulfone with 5% sulfonated PEEK. During a dead-end filtration test, we found that a SPEEK-PSf membrane could reject or adsorb the vast majority of cyclic MC-LR and virtually all linearized MC-LR. The partial success of modified membranes and reactive processes suggests that a physical barrier membrane with additional functionality (e.g., charge, reactivity) may result in enhanced MC-LR remediation. The SPEEK-PSf membrane was effective at rejecting or adsorbing 97% cyclic MC-LR and virtually all linear MC-LR. MC-LR was found to reversibly adsorb to the SPEEK-PSf membranes, with desorption of MC-LR occurring when the membranes were soaked in a methanol/water solution. This is highly advantageous because it provides a method of removing MC-LR from feed waters via reversible adsorption to membranes that can then be cleaned and reused.

## 4. Materials and Methods

### 4.1. Chemicals

Polysulfone (PSf) with a molecular weight of 35,000 g/moL, sodium phosphate monobasic, and sodium phosphate dibasic were obtained from Sigma Aldrich (St. Louis, MO, USA). ACS Grades N-Methyl-2-pyrrolidone (NMP), BDH1141-4LP, 95–98% concentrated sulfuric acid(H_2_SO_4_), BDH3070-25LPC, methanol, and acetic acid (glacial) were obtained from VWR BDH Chemicals (Solon, OH, USA). Polyether ether ketone (PEEK) pallets, 23969-50, were obtained from Polysciences Inc., (Warrington, PA, USA). Microcystin-LR was obtained from Cayman Chemical (Ann Arbor, MI, USA).

### 4.2. MlrA Expression and Experiments

#### 4.2.1. Strain and Plasmid Construction

*E. coli* chemically competent BL21 (DE3) was purchased from New England Biolabs Inc. and used as a host to produce targeted recombinant protein. the pET-21a (+) plasmid was purchased from Novagen Life Technologies (Millipore Sigma, Burlington, MA, North America). A DNA fragment encoding to MlrA was designed and purchased from Integrated DNA Technologies (Coralville, IA, USA). After plasmid construction, Recombinant pET-MlrA plasmid was transformed into *E. coli* Bl-21 cells by heat-shock method.

#### 4.2.2. Media, Growth, and Expression of Recombinant MlrA Enzyme

Luria-Bertani (LB) broth containing ampicillin was used for all plates and seed cultivations. A single colony of *E. coli* BL21 containing cloned pETmlrA plasmid picked up from agar plate was used to inoculate 10 mL of LB media supplied with 75 μg/mL ampicillin as an initial growth seed culture. The culture was incubated at 37 °C with shaking at 200 rpm overnight. This experiment was performed by using 2.5 mL of overnight growth to inoculated 1 L supplemented with 75 μg/mL ampicillin. The growth was incubated at 37 °C with 200 rpm shaking speed. Isopropyl β-D-1-thiogalactopyranoside (IPTG) with final concentration of 0.5 mM was used to induce recombinant plasmid at the mid-exponential phase of growth when the optical density 1 reached ~0.6 units (at 600 nm). After 4 h from induction, cells were harvested to proceed with further experiments.

#### 4.2.3. Cell Lysate Preparation and Hisx6-tag MlrA Purification

After expression was completed, cells were harvested by centrifugation at 4500× *g* for 45 min. Cell pellets were re-suspended in 10 mM sodium phosphate buffer, pH 7.4, and subjected to sonication on ice with a Qsonica sonicator on a 20 s burst cycle (power 10). The cell homogenate was centrifuged to clarify the cell lysate. Supernatants were collected to proceed with further experiments, such as testing for enzyme activity and purification. The fast protein liquid chromatography (FPLC) system from ÄKTA Amersham Pharmacia Biotech was used to purify the cell lysate. Hisx6-tag was designed to be in the C-terminal of MlrA enzyme to facilitate purification through Immobilized Metal Affinity Chromatography (IMAC). MlrA is a putative metallopeptidase, and it has been shown that nickel ions inhibit metalloproteases through substitution of the active center ion [28]. Even though Ni-NTA is an efficient binding to His6x-tag proteins [29], cobalt-loaded HiTrap IMAC FF column (Co-NTA) was used to purify expressed protein. Sodium dodecyl polyacrylamide gel electrophoresis (SDS-PAGE) and Western blot using anti-Hisx6-tag antibody (abcam, USA) was used to ensure that the targeted protein expressed and produced properly by the host and to confirm the purity.

#### 4.2.4. Microcystin-LR Linearization Activity of MlrA

The MlrA cyanotoxin linearization activity was tested by incubating MlrA with microcystin-LR. Two mL solution of 100 µg/L MC-LR and 216 mg/L MlrA in phosphate buffer (5 mM) was incubated in a glass scintillation vial for 24 h at room temperature. Samples of 500 µL were taken at 0, 4, and 24 h after initial mixing and transferred to glass HPLC vials. The reaction in each HPLC vial was stopped by 50 µL of 5% acetic acid. Samples were stored at 4 °C until analysis. The linearization of MC-LR was monitored by LC-MS.

#### 4.2.5. LC-MS Methods for Detecting MC-LR

Cyclic and linear MC-LR were detected using ultra performance liquid chromatography-tandem mass spectrometry (UPLC-MS/MS). A LCMS-8040 Triple Quadrupole Liquid Chromatograph Mass Spectrometer (Shimadzu, Columbia, MD, USA) was used to analyze the samples. A Shimadzu C18 column (2.1 × 50 mm, 1.9 um particle size) was used for all samples except for those from membrane filtration experiments (Section 4.3.5), for which a Waters Acquity UPLC peptide HSS T3 column (1 × 100 mm, 1.8 um particle size) was used. An injection volume of 10 μL was used, and the column was held at 40 °C. The mobile phases were water with 0.1% formic acid and acetonitrile with 0.1% formic acid. The series of gradients were as follows: the column was initially balanced with 20% acetonitrile for 1 min, then increased from 20% to 80% acetonitrile over 7 min, then held at 80% acetonitrile for 1 min, and then returned to 20% acetonitrile and held for 1 min.

For mass spectrometry, electrospray ionization was used. Full scans were performed for *m*/*z* 400–1200 in positive ion mode. Additionally, single-ion monitoring (SIM) was used to detect peaks at *m*/*z* = 498.5 and 995.7 (2 + and 1 + charge states for the cyclic MC-LR) and for *m*/*z* = 507.4 and 1013.5 (2 + and 1 + charge states for linear MC-LR). Percent change in concentration of cyclic MC-LR was calculated by the relative change in the area under the peaks for cyclic MC-LR in the SIM chromatograms.

### 4.3. Membrane Fabrication and Experiments

#### 4.3.1. Sulfonation of SPEEK

The sulfonation procedure was adapted from literature [23] and modified as discussed here. In the method, 25 g of PEEK granules were heated in a vacuum oven at 100 °C for 24 h. They were then crushed into fine powder, which was dissolved in concentrated sulfuric acid in a 90:10% (H_2_SO_4_:PEEK) at room temperature for 48 h, then precipitated out of the solution using cold deionized water. The precipitant was neutralized and dried overnight in a vacuum oven at 100 °C.

#### 4.3.2. Membrane Formation

The membranes were formed via non-solvent induced phase separation (NIPS). In this method, once the membrane is cast, it is immersed into a non-solvent, where a phase transition takes place [30]. The membrane formed contains a polymer-rich surface (active rea) and a polymer poor-pore structure suitable for ultrafiltration [30,31]. The non-solvent utilized was water. The dope solution was prepared by dissolving PSf blended with SPEEK (95:5%) into N-Methyl-2-pyrrolidone solvent to make 79:21% NMP:SPEEK-PSf. The blended dope solution was then spread onto a glass plate using a doctor blade (0.4 mm) and exposed to air for 15 s before being immersed in deionized water. The formed membrane was then stored in deionized water [23].

#### 4.3.3. Membrane Characterization

To understand the structural morphology, functionality, and elemental composition of the membranes, various characterization techniques were utilized, such as scanning electron microscopy (SEM) for morphology, Fourier-transform infrared spectroscopy (FT-IR), and X-ray photoelectron spectroscopy (XPS) for structural and contact angle measurements for hydrophobicity.

Surface and cross-sectional images of the membrane were obtained using an FEI Quanta scanning (Hillsboro, OR, USA) scanning electron microscope. To avoid charging thermal decomposition of the membrane samples, the surface of the membranes was coated with palladium using Leica ACE 600 carbon/sputter coater (Fremont, CA, USA). To confirm the elemental composition within the as synthesized membranes, a K-Alpha X-ray Photoelectron Spectroscope (Waltham, MA, USA) was utilized. Membrane preparation involved freeze drying of samples with liquid nitrogen, followed by data analysis using Fisher Scientific Avantage software. A NEXUS 470/670/870, 110 W, 5 V/12 V ESD (Thermo Nicolet, Madison, WI, USA) Fourier-transform infrared spectrometer (FT-IR) was utilized to identify and confirm the functional groups present within the membrane. Lastly, the hydrophilicity/hydrophobicity nature of the membrane was determined via the water contact angle measurements using a drop shape analyzer (Kruss DSA100, Matthews, NC, USA) to determine the average contact angle of the membrane samples.

#### 4.3.4. Adsorption and Desorption of Microcystin-LR on Membrane

The adsorption and desorption of MC-LR was tested by incubating MC-LR with a SPEEK-PSf membrane to adsorb and then adding methanol to desorb. A one-fourth section of a SPEEK-PSf membrane (~4.33 cm^2^) was incubated with 100 µg/L MC-LR in phosphate buffer (5 mM) in a glass scintillation vial for 24 h at room temperature. Samples of 500 µL were taken at 0, 4, and 24 h after initial mixing and transferred to glass HPLC vials. Before taking the sample at 24 h, 667 µL methanol was added to the scintillation vial (bringing the concentration of methanol to 40 v%) and swirled to mix for 5 min. Samples were combined with 50 µL of 5% acetic acid to maintain consistency with other tests. Samples were then stored at 4 °C and analyzed by LC-MS.

#### 4.3.5. Membrane Filtration

The byproducts of enzymatic linearization of MC-LR were filtered through the SPEEK-PSf membranes using a dead-end filtration cell from Millipore^®^, Burlington, MA, USA, with max operating pressure 5 bar. To provide the required pressure drop for flow, nitrogen gas was employed under a constant pressure of 4.1 bars (60 psi). Pre-compaction of the membrane with deionized water was carried out, followed by filtration of the byproducts at the end of the 24 h linearization study. The filtrate after MC-LR linearization, which was the membrane feed solution, included linearized byproducts, potentially MC-LR, methanol, phosphate buffer (containing disodium hydrogen phosphate and potassium phosphate dibasic), acetic acid, and MlrA enzyme solution (Table 4). The reaction volume was concentrated by four times for all components while maintaining the same reactant concentration ratios. Samples were analyzed by LC-MS.

## Figures and Tables

**Figure 1 toxins-14-00231-f001:**
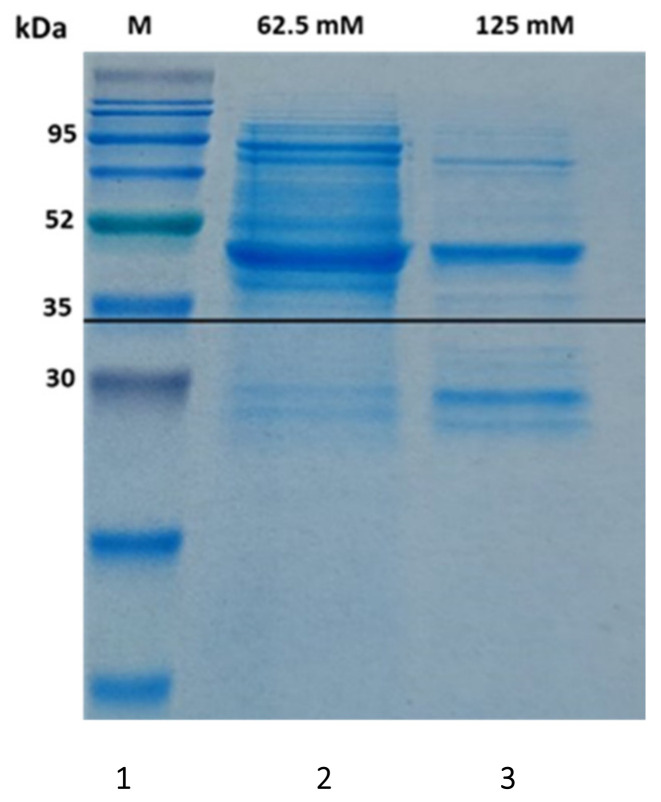
SDS-PAGE stained with Coomassie Brilliant Blue. (**1**) molecular weight marker; (**2**) MlrA eluted with 62.5 mM imidazole; (**3**) MlrA eluted in 125 mM imidazole.

**Figure 2 toxins-14-00231-f002:**
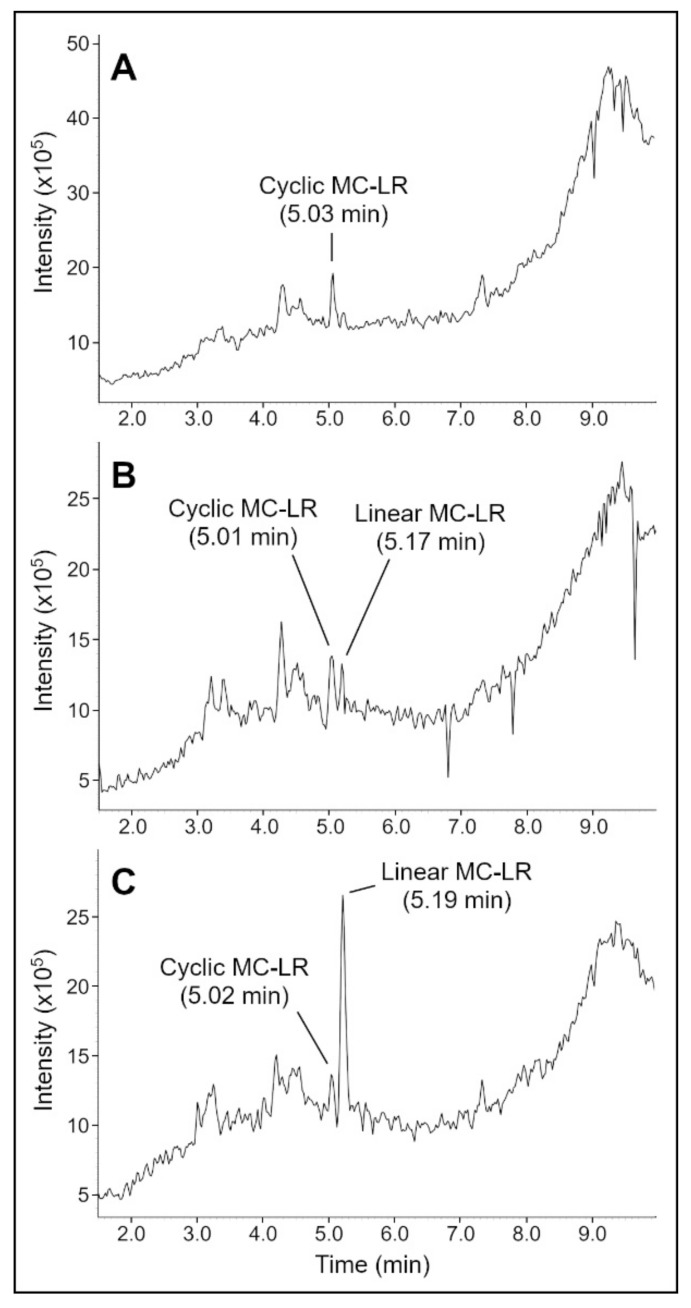
LC-MS chromatograms of the MC-LR as it was linearized by MlrA in an environment of phosphate buffer (5 mM) for samples taken 0 h (**A**), 4 h (**B**), and 24 h (**C**) after initial mixing.

**Figure 3 toxins-14-00231-f003:**
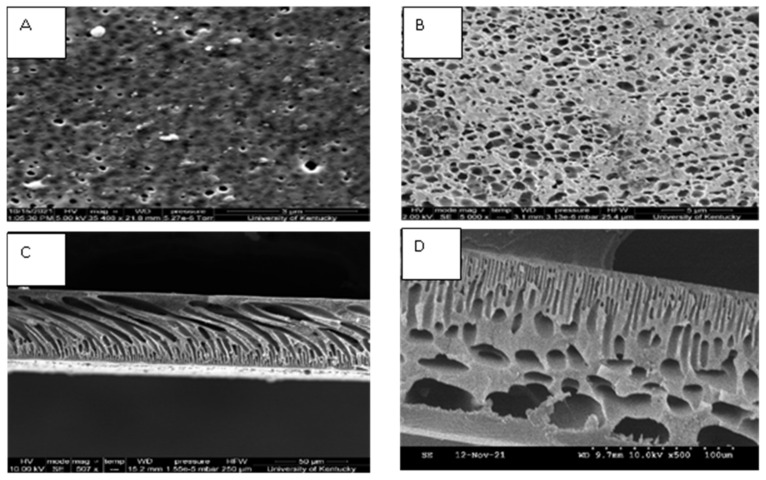
SEM images showing the surface morphology and cross section of SPEEK-PSf membranes before and after filtration: (**A**) surface image before filtration, (**B**) surface image after filtration, (**C**) cross-section image before filtration, and (**D**) cross-section image after filtration.

**Figure 4 toxins-14-00231-f004:**
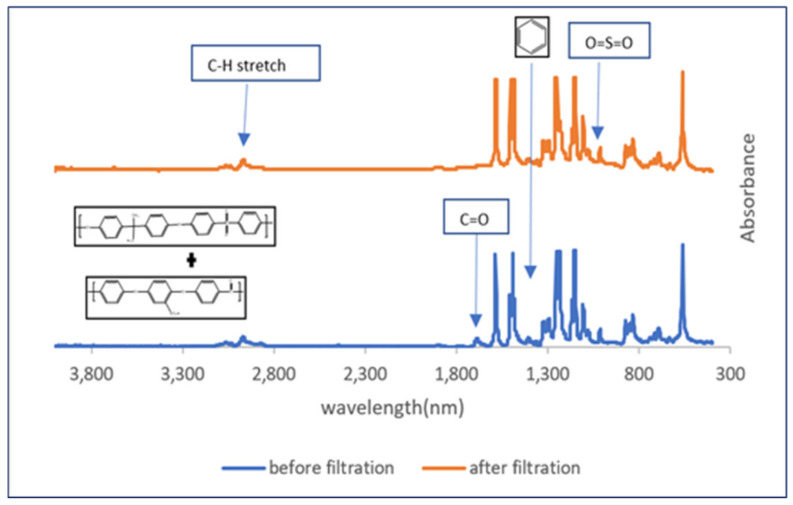
FT-IR spectra of SPEEK-PSf membrane before and after filtration of the linearization study byproducts.

**Figure 5 toxins-14-00231-f005:**
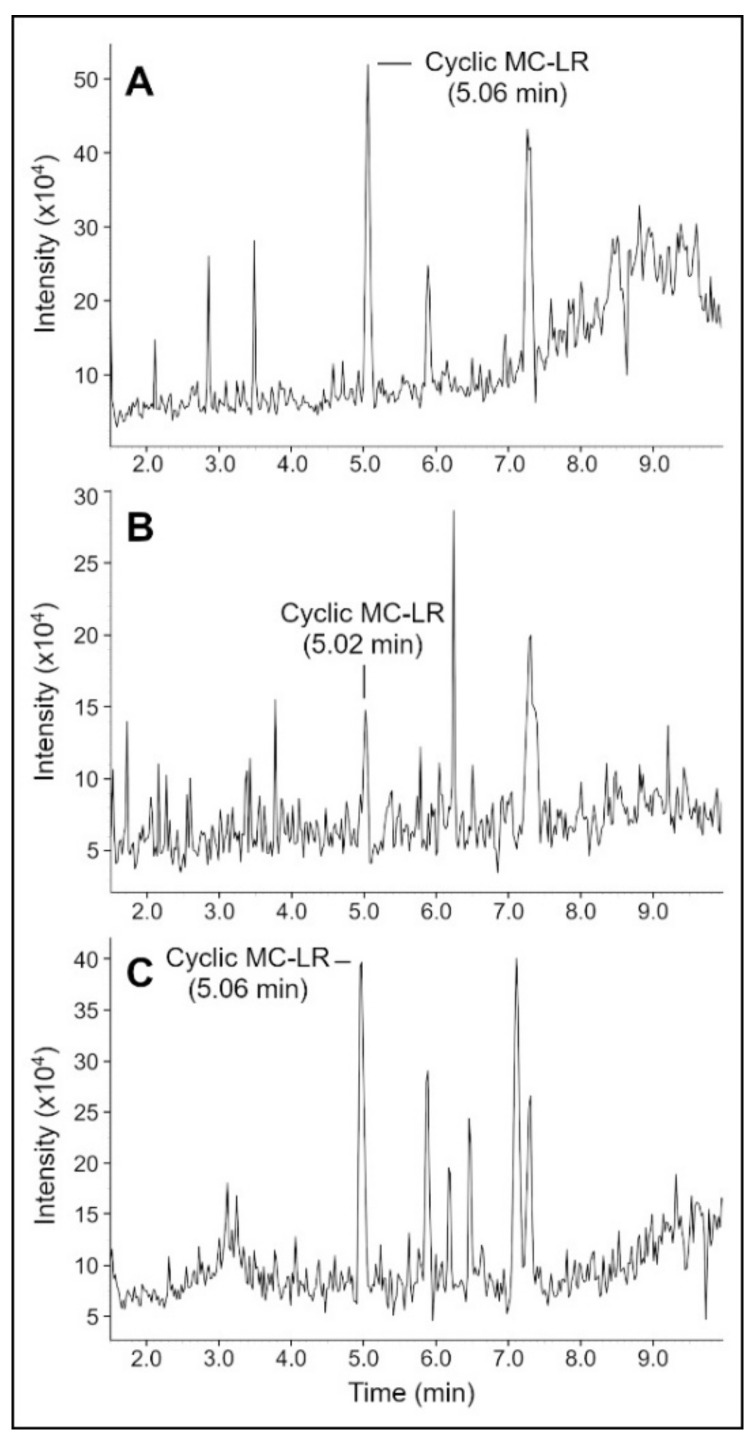
LC-MS chromatograms of the MC-LR solution incubated with a SPEEK-PSf membrane for samples taken at 0 h (**A**), 4 h (**B**), and 24 h (**C**) after initial mixing. Note that the sample taken at 24 h was mixed with methanol to desorb MC-LR from the membrane.

**Figure 6 toxins-14-00231-f006:**
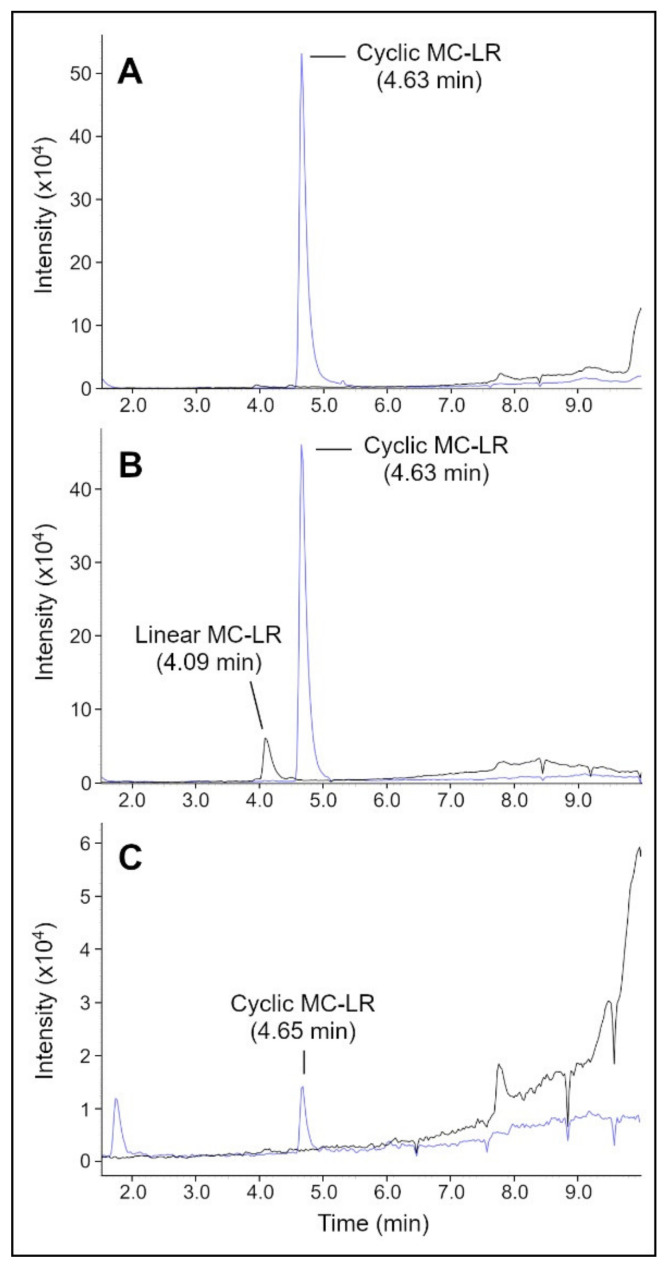
LC-MS chromatograms of the MC-LR solution incubated with MlrA for samples taken at 0 h (**A**), and 24 h (**B**) after initial mixing and for the solution after 24 h filtered through a SPEEK-PSf membrane (**C**). Graphs for the cyclic MC-LR region (*m*/*z* = 498.5) are in blue, while graphs for the linear MC-LR region (*m*/*z* = 507.4) are in black.

**Table 1 toxins-14-00231-t001:** XPS spectrum of SPEEK-PSf membrane after filtration.

Name	Peak BE	FWHM eV	Area (*p*) CP	Atomic%
O1s	530.75	3.14	981,557.4	20.25
C1s	284.11	3.57	1,041,084	51.95
N1s	398.37	1.69	295,204.8	9.49
Na1s	1070.11	2.75	101,255.2	1.03
Cl2p	196.83	3.43	34,208.35	0.59
S2p	166.82	2.96	33,040.04	0.82
P2p	132.07	3.01	19,959.85	0.67
Ca2p	346.13	2.43	28,255.31	0.25
Si2p	100.88	3.68	13,253.4	0.66

**Table 2 toxins-14-00231-t002:** XPS spectrum of SPEEK-PSf membrane before filtration.

Name	Peak BE	FWHM eV	Area (*p*) CP	Atomic%
C1s	284.92	3.11	1,420,861	71.25
O1s	532.15	3.06	1,064,064	22.08
S2p	167.91	2.69	77,509.07	1.92
N1s	399.44	1.71	109,829.3	3.55
Na1s	1071.35	2.82	82,496.91	0.85
Cl2p	198.08	5.38	20,673.64	0.36

**Table 3 toxins-14-00231-t003:** TOC measurements showing the amount of total dissolved carbon in the MC-LR stock solution before enzyme linearization, before membrane filtration, and after membrane filtration.

Solution	TOC Measurements (mg/L)	Components
MC-LR stock solution before linearization	252	MC-LR, water, ethanol, phosphate buffer, and acetic acid
After linearization before filtration	396	MC-LR, water, ethanol, acetone, MlrA enzyme, and acetic acid
After linearization after filtration	366	MC-LR, water, ethanol, acetone, MlrA enzyme, and acetic acid

**Table 4 toxins-14-00231-t004:** Chemicals present in the stock MC-LR solution at t = 0 s, solution before linearization, and solution after linearization.

Chemicals	Concentration (mg/L)	Volume Added to Stock Solution (µL)	Mass That Should Be Present (mg)	Volume Present in Solution after Linearization	Mass That Should Be Present (mg)
MC-LR in phosphate buffer	1	720	0.72	720	0.72
Phosphate buffer 5 mM	2.06	1800	3.69	6480	13.32
MlrA solution	2.4	0	0	800	1.92
Acetic acid	50	200	10	200	10
Potassium phosphate monobasic	25.4	1.28	0.03	1.28	0.03
Potassium phosphate dibasic	34.1	0.95	0.03	0.95	0.03

## Data Availability

The data presented in this study are available on request from the corresponding author.

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
