# Peer review of "Microcystin-LR Removal from Water via Enzymatic Linearization and Ultrafiltration"

_toxins, 2022, doi:10.3390/toxins14040231_

Round 1

Reviewer 1 Report

Review: Microcystin-LR destruction and removal from water via enzy-2 matic biodegradation and ultrafiltration

Submitted to Toxins, Feb 2022

The authors examined using microcystinase A (MlrA) and membrane filtration to remove microcystin LR, a cyanotoxin that is common in freshwater system. I am a microbiologist/chemist, I am not very familiar with the membrane material modification part. Therefore, I don’t feel qualified on commenting on that part.

My major concern/question is I am not convinced that this is “a novel strategy of remediation of microcystin-tainted water, combining degradation of the toxin with removal by membrane filtration”.

  1. The Mlr degradation of MC-LR were done in “a glass scintillation vial”. The volume of this experiment was too small to make it applicable to scale up to environment-relevant or even drinking water-use scope.
  2. The Mlr degradation and membrane filtration were done at different step and the Mlr degradation essentially failed. The point of including Mlr degradation was not clear.
  3. The data presentation was not sufficient. LC/MS results were only shown as chromatographs. And there is no quantification data. Depending on how stable the instrument is, sometime such figure could give misleading results. Figure 2, 5 had messy baselines. I think for publication purpose, quantification data should be presented.
  4. The filtration step was done at 5 bar (>60Psi) with N2 No volume was provided. Again, I am not sure this is an applicable approach for MC-LR removal.

Other comments

The result section contains a lot of “present” tense sentences, this is a grammar issue.

Reviewer 2 Report

This study describes a strategy for the remediation of microcystin-LR with a membrane filtration. This study is useful and a good contribution in the field. However, some points have to be considered as including some analytical parameters to be revised for a better evaluation of the degradation process.   

  1. The discussion focuses mainly on the results, an interpretation of these results on the applicability of this method according to the results obtained should be included.
  2. Only MC-LR was considered in this study when it is known that other congeners can be predominant over LR such as LA or RR, and multiple other congeners (YR, WR, LY, LF, etc) and can be present at high concentrations in toxic cyanobacterial blooms. The use of this remediation method on other microcystins should be mentionned in the discussion as a perspective for further research or use.
  3. Line 125 and further : m/z must be in italic.
  4. Section 2.2.5: The manuscript indicates that there will be a search for degradation products but only the linear MC-LR is considered. Many other major degradation products appearing from biodegradation processes can also be detected and should be included in this study to better understand the loss of MC-LR.
  5. Line 226: It is the sensitivity that is increased with SIM mode. Moreover, a fragmentation mode should be used for an even better sensitivity and selectivity in detection since a triple quadrupole mass spectrometer was used for this study.
  6. Figure 2, 5 and 6: In this study, and in these figures, TIC (total ion chromatograms) are used to evaluate the MC-LR signals. However, it is extracted chromatograms that should be used to better see the difference in peak intensity. The extracted chromatograms should be generated to eliminate de baseline and have a clean peak intensities and area. In this case, the peak area should be used for a more precise evaluation of the MC-LR loss in the degradation process.
  7. Line 352 : please specify that columns were used for chromatographic separation.
  8. Line 247 (same as Comment 6): Extracted ion chromatograms is used to extract targeted peaks from background so it can be used for qualitative and quantitative purpose. The use of TIC for peak evaluation is not proper for a precise analysis.

Reviewer 3 Report

The authors describe a method for MC-LR remediation that includes linearization of the toxin utilizing the microcystinase A enzyme with membrane filtration rejection of linearized byproducts. The authors show that MlrA was successful in linearizing around 86 percent of the MC-LR over 24 hours using techniques like affinity chromatography, membrane adsorption/desorption, and LC-MS. Although I do not see much novelty in the approach, as it seems obvious to do so, I believe that this study certainly adds up to the effort that advances the field. The manuscript contains several grammatical errors that are hard to summarize in this report. Therefore, I urge the authors to resolve them. I do not see many issues with the data, but the language makes it hard to read and even confuses the reader at times. The manuscript can be accepted if the presentation and language can be improved to the standard of standard scientific literature.

Round 2

Reviewer 1 Report

there are still a few minor grammar issues needs to be fixed, for example

in abstract, "The MlrA enzyme was overexpressed in E. coli" change overexpressed to expressed. There is no comparison overexpress is a wrong word. Also a full name of E coli needs to be present afterwards, it can be abbreviated. 

In method, sentence can't start with a number, it needs to be spelled out if you have to start a sentence with a number

Reviewer 2 Report

The authors have responded adequately to the comments of the first revision.

Author Response

We revised the manuscript and responded to first round of revisions.